# Proactive Threat Hunting in Critical Infrastructure Protection through Hybrid Machine Learning Algorithm Application

**DOI:** 10.3390/s24154888

**Published:** 2024-07-27

**Authors:** Ali Shan, Seunghwan Myeong

**Affiliations:** 1Center of Security Convergence & eGovernance, Inha University, Incheon 22212, Republic of Korea; kralishan@gmail.com; 2Department of Public Administration, Inha University, Incheon 22212, Republic of Korea

**Keywords:** cyber-attacks, threats, machine learning, cyber-security

## Abstract

Cyber-security challenges are growing globally and are specifically targeting critical infrastructure. Conventional countermeasure practices are insufficient to provide proactive threat hunting. In this study, random forest (RF), support vector machine (SVM), multi-layer perceptron (MLP), AdaBoost, and hybrid models were applied for proactive threat hunting. By automating detection, the hybrid machine learning-based method improves threat hunting and frees up time to concentrate on high-risk warnings. These models are implemented on approach devices, access, and principal servers. The efficacy of several models, including hybrid approaches, is assessed. The findings of these studies are that the AdaBoost model provides the highest efficiency, with a 0.98 ROC area and 95.7% accuracy, detecting 146 threats with 29 false positives. Similarly, the random forest model achieved a 0.98 area under the ROC curve and a 95% overall accuracy, accurately identifying 132 threats and reducing false positives to 31. The hybrid model exhibited promise with a 0.89 ROC area and 94.9% accuracy, though it requires further refinement to lower its false positive rate. This research emphasizes the role of machine learning in improving cyber-security, particularly for critical infrastructure. Advanced ML techniques enhance threat detection and response times, and their continuous learning ability ensures adaptability to new threats.

## 1. Introduction

In the modern world, sharing of information and other professional operations like business, commerce, bank transactions, advertising and services are becoming more prevalent owing to cyber-civilization [1]. However, cybercriminal activities have also increased exponentially in cycle with the exponential growth in the usage of cyberspace [2]. The primary cause of this rise is the widespread overuse of Web apps in practically every aspect of daily life. These Web apps have some design flaws, which cybercriminals take advantage of to obtain unauthorized access to the networks [3]. Traditional security solutions only respond to known dangers, and they are ill equipped to successfully tackle emerging and unpredictable cyber-threats. These solutions ignore insider threats and sophisticated attacks, so relying on them is insufficient to safeguard enterprises handling sensitive data [4,5]. Modern digital ecosystems thus necessitate a break from traditional security concepts and incorporation of cutting-edge techniques and technology [6]. Therefore, cyber-security research area has become an important concern for researchers and practitioners. In order to combat untraceable software attacks, the threat hunters’ assistance is very crucial in this situation [7]. Therefore, hunting is performed with the aim of protecting the organization prior to the occurrence of an attack. There are two types of threat hunting: proactive and reactive. In proactive, threat hunting eliminates all potential avenues of attack while simultaneously looking for possibly harmful activity, whereas reactive hunting responds to potentially malevolent behavior in attack hypotheses [8,9]. Machine learning is an effective proactive approach, used to analyze large datasets in real-time and spot unusual patterns and behaviors [10]. By innovations of these technologies, cyber-defense gains a proactive component, replacing reactive tactics that find it difficult to keep up with the constantly shifting threat landscape [11]. To combat diverse security breaches, a variety of research initiatives have been performed in distinct cyber-sectors, each with unique features and characteristics. To facilitate infrastructure security, this research attempts to implement a proactive strategy using machine learning. Assessments were conducted on the effectiveness of many sophisticated machine learning models, including random forest, support vector machine, hybrid machine learning, AdaBoost, and multi-layer perception models. To find the optimal model for threat hunting, comparisons between various models were also made. Furthermore, these models continuously learn from new incoming threats, improving the model accuracy and performance with respect to time.

### Research Gap

It is evident from previous studies that a significant amount of study has been conducted in the fields of cyber-security. Methodological mapping investigations of SLRs have also been carried out. However, most of the mapping studies that are now available are about cyber-attack mitigation through different mathematical techniques. There is no systematic mapping research that compiles information on proactive cyber-security attack measurements through hybrid machine learning. This mapping project is being carried out to close the gap by giving researchers a general understanding of the current cyber-security vulnerabilities and methods for detection and mitigation. To fill these research gaps, this study focused on the following research questions:

RQ1: How well do the machine learning models handle real-time data from critical infrastructure sources?

RQ2: How successful are the individual machine learning algorithms (RF, SVM, MLP, AdaBoost) in identifying and mitigating threats in CI protection?

RQ3: How accurate and effective is the hybrid ML model at detecting threats compared to individual models?

RQ4: How do models adapt to new and evolving threats when processing real-time data for threat hunting?

RQ5: Which model is found to be the best optimized in terms of testing and validation strategies?

These research questions aim to critically explore the multifaceted aspects of implementing machine learning models as a proactive approach for threat hunting in critical infrastructure protection. The main contributions of this study are as follows:Comprehensive analysis of the role of various techniques including proactive, mathematical, machine learning, and hybrid strategies for threat detection.Development of machine learning models (RF, SVM, MLP, AdaBoost, and hybrid) to increase the attack detection accuracy and robustness in critical infrastructure.Comparison of the effectiveness of all models using ROC, precision, recall, accuracy, F1-score, and learning curves. Development of the most optimized models that deal with real-world scenarios to detect cyber-attacks with reduced false situations.

The rest of the paper includes Section 2, which comprises a brief literature review regarding cyber-security, mathematics, machine learning, and hybrid solutions as proactive approaches for intrusion detection. Section 3 describes the general methodology considered in this work. Section 4 includes the results, discussion, and comparative analysis, followed by concluding remarks. 

## 2. Literature Review

### 2.1. Cyber-Security

“Cyber-security defined as the protection against unwanted attacks, harm, or transformation of data in a system and also the safety of systems themselves” [12]. It is concerned with the security and privacy of digital assets, including networks, computers, and data that are processed, stored, and transferred via Internet-based information systems, according to ISACA. The Worldwide Telecommunications Union defines cyber-security as the set of methods, guidelines, protocols, best practices, and procedures used to safeguard users’ online assets and organizations [13]. Cyber-security, according to the Merriam-Webster definition, is the defense of computer systems against intrusions and illegal access [14,15]. Cyber-security comprises the techniques and equipment used to defend computer networks and devices from assaults and illegal access over the Internet. Cyber-security is the defense against unauthorized access to an organization’s non-physical and physical components. Diverse definitions among scholars indicate how they define cyber-security [16]. The current definitions concentrate on several facets of cyber-security. Several definitions, for instance, emphasize privacy and protection, while others emphasize the necessity of establishing guidelines and procedures for availability, confidentiality, and information integrity. Cyber-security may be viewed as a defense against unwanted access to the assets of people and organizations. The significance of the cyber-ecosystem and its preservation is further emphasized by these concepts [17].

### 2.2. Cyber-Security Terminologies

There are certain terminologies that are required to follow the main ideas associated with the field of study. 

**Cyberspace:** The use of the digital and electromagnetic fields to create, update, store, share, and exploit details with the aid of interrelated and reliant networks using the most recent information and communication methods is the distinctive feature of cyberspace, a global domain inside the information world [18,19].**Vulnerabilities:** These are the weaknesses in a system’s architecture that provide an attacker with the ability to carry out malicious instructions, gain unauthorized access to data, and/or launch different types of denial-of-service assaults [17,20].**Threats:** These are the kinds of things done to take advantage of security weaknesses in an infrastructure and make it worse [21].**Attacks:** These are the steps taken to use different tools and techniques to attack vulnerabilities in a system to harm it or interfere with its regular operations. Attackers carry out these attacks to fulfill their malevolent objectives, which may include monetary gain or self-gratification [22].

A multitude of security flaws have been covered in the literature. The following is a description of several typical cyber-security vulnerabilities to help the readers better understand them:**Denial-of-service (DoS):** The goal of this kind of attack is to prevent authorized users from accessing a computer or network resource [23]. Any incident that reduces or removes a network’s ability to carry out its intended function is the root cause of it. Most computer devices on the Internet of Things (IoT) ecosystem are susceptible to asset enervation attacks because of their tiny memory capacities and constrained computational resources [24]. Potential attackers make use of the fact that different businesses employ similar technologies, which is one of the reasons why denial-of-service attacks occur [25].**Malware:** In this attack, the attacker takes advantage of security flaws in computer systems to install malicious software and obtain unauthorized access. A significant financial or political payoff is the driving force behind malware, encouraging attackers to infiltrate as many network devices as they can to further their malevolent objectives [26].**Phishing:** This is a criminal activity that gathers private information from Internet users by using social engineering and technology. Phishing strategies use a variety of communication channels, including pop-up windows, instant chats, email, and websites [27,28].**SQL injection attack:** To alter or manipulate an SQL query to the attacker’s benefit, an input string is introduced through the application in this attack. The database is harmed by this assault in several ways, including sensitive data exposure and unauthorized access and modification [29]. This assault is dangerous since it has the potential to disrupt functionality and secrecy through data loss or unauthorized organizations misusing the data. Moreover, this type of assault also executes orders at the system level, which prevents authorized users from gaining access to the necessary data [30].

### 2.3. Proactive Approach

Modern computer systems often contain information that is highly valuable to competitors, foreign governments, or criminal entities. As these systems increasingly interconnect, the likelihood of attacks by these adversaries also escalates. As a result, many enterprise networks currently find themselves under cyber-attack or have been attacked in the past [31,32]. This has led to a significant growth in the security tools market, aimed at defending systems and identifying attacks. Nonetheless, many of these tools operate on specific, predefined logic, such as monitoring certain gateways for threats, which limits their scope [33]. In this context, the security functions within organizations focus on detecting active threats, a method typically referred to as a “Reactive Approach.” This strategy centers on recognizing and addressing actions that adversaries have initiated or are in the process of executing [34]. Some of the important proactive techniques used in literature are given in Table 1.

Critical infrastructure includes both physical and cyber-systems that are necessary for a society’s basic functions and security [42]. These systems include those related to electricity, water, transportation, telecommunications, and healthcare. Vulnerabilities in the energy industry result from possible assaults on infrastructure used for the production and distribution of power [43]. Cyber-attacks targeting water treatment controls and system pollution are two problems that might affect water infrastructure [44]. Physical structural disturbances or cyber-attacks targeting operating systems can have a significant impact on transportation networks, encompassing seaports and airports. Infrastructure related to telecommunications, which is necessary for emergency response and communication, is vulnerable to both physical damage and cyber-attacks. Cyber-attacks that aim to compromise sensitive data and cause service interruptions are especially dangerous for the healthcare industry, the vitality of this infrastructure, and its growing dependence [45].

### 2.4. Mathematical Techniques and Machine Learning

In many scientific fields, mathematical models are commonplace and essential for solving problems [46]. They provide a summarized mathematical depiction of reality that aids in understanding intricate systems, resolving problems, and obtaining vital information for well-informed decision-making. These models use algorithms to determine which solution best fits the given problem. Within the mathematical sciences, computational optimization is a well-known topic that focuses on determining a function’s extreme values, such as its maximum or lowest value [33,47]. Since they give a variety of systems a competitive edge, optimization techniques have grown in importance over the past 20 years for management, decision-making, technology advancement, and general development. Numerous methods have been developed to solve problems in optimization and machine learning due to the practical importance of both domains [48]. None of those algorithms are perfect, even though most of them work well to solve the problems they are applied to. Many constraints related to optimization, as well as machine learning techniques, are well documented in the literature. Malware categorization issues are related to attack-detection issues [49]. Machine learning methods are used to classify binary data as benign or malicious. Support vector machine (SVM), K-nearest neighbor (KNN), and decision tree (DT) models are supervised non-parametric methods, whereas naïve Bayes (NB) is a parametric algorithm [50,51]. By dividing the feature set using splitting functions, the DT algorithm produces a decision tree. Two crucial DT algorithm parameters, tree depth and splitting criteria, are adjusted to create the classifier [52]. In the KNN algorithm, the prediction of input instance is conducted based on the nearest k-training instances [53]. The distance metric and the value of k, which represents the nearest k instances, are essential parameters in classifier training. Support vector machine (SVM) finds a hyperplane which can classify the dataset into a different class. The SVM classifier is trained with two essential parameters: the kernel function and hyperparameter (c value), which plays a vital role in creating the hyperplane for separating the non-linear feature space into different classes [54,55]. The ensemble machine learning algorithms random forest (RF), AdaBoost, and gradient boosting (GB) build multiple weak classifiers and then build a strong classifier. Some ensemble algorithms provide an option to build weak learners using different simple algorithms like SVM and KNN. Each ensemble machine learning algorithm uses different science and mathematical equations to develop the final classifier. The only difference is when applying them to the training dataset [56].

### 2.5. Hybrid Techniques

A hybrid algorithm that blends optimization and machine learning techniques utilizes the strengths of both to establish a robust framework for addressing complex challenges [57]. This approach improves decision-making by weaving optimization methods into the machine learning process and vice versa. The result is that optimization helps to steer the learning process, thus enhancing both the precision and efficiency of decisions. This synergistic combination leverages both explicit mathematical optimization and data-driven learning capabilities, leading to more effective and streamlined decision-making [58]. In the field of cyber-security, hybrid machine learning approaches are increasingly employed to strengthen defenses against a wide range of cyber-attacks. These hybrid methods blend different machine learning techniques to enhance threat detection, response capabilities, and predictive accuracy [59]. For instance, anomaly detection algorithms learn normal network or system behavior to detect both known threats and unusual activities that could signify new attacks when combined with signature-based methods. Predictive analytics integrated with threat intelligence platforms use historical data and current trends to forecast attack vectors and potential targets. Ensemble learning techniques improve threat classification by combining multiple classifiers, while deep learning models combined with rule-based systems enforce specific security policies and checks [60]. Adaptive systems utilizing reinforcement learning continually refine strategies for detecting and mitigating evolving threats. Additionally, neuro-fuzzy systems help interpret the significance of network anomalies in ambiguous or incomplete data. These hybrid models enable cyber-security systems to process vast and varied datasets, recognize complex patterns, adapt to new threats, and enforce security protocols efficiently, resulting in more robust and intelligent cyber-security solutions [61]. Different business companies depend upon cyber-security experts that are known as threat hunters. These security experts defend all types of cyber-attacks in a timely manner, even zero-day attacks, with real-time data [62]. To improve business security, most organizations base their systems on artificial intelligence. Although different types of machine learning model are used for cyber-security, not all of these models are used for proactive techniques based on real-time data [63]. 

### 2.6. Significance of Machine Learning

As cyber-threats become increasingly diverse, sophisticated, and targeted, automation is emerging as a crucial tool for security teams, which are often overwhelmed. Today’s security landscape is fraught with challenges such as malware, phishing, ransomware, denial-of-service (DoS) attacks, and zero-day exploits. Traditional defense mechanisms are imperfect, and many detection methods still depend heavily on manual analysis and decision-making by security analysts to identify advanced threats, malicious behaviors, and significant risks. Machine learning excels in identifying and predicting specific patterns, outperforming human capabilities in this area. However, security decisions and policy adjustments often struggle to keep pace with the demands of complex and dynamic network environments. Leveraging machine learning for intelligent decision-making, automation in the realm of cyber-security has become increasingly feasible. This study also aimed to identify and comparatively assess the effectiveness of ML models. Various models, including RF, SVM, MLP, AdaBoost, and hybrid models, were comprehensively assessed for threat hunting in real-time cyber-security. All models performed well with high accuracy and precision in identifying anomalies, except for MLP, compared to various previously described models [41,64,65,66,67]. Previously reported SVM, RF, NB, and logistic regression models exhibited 94% accuracy for threat detection [51]. Similarly, a CNN analyzed for intrusion detection in an IoT system showed maximum accuracy of 93% [53]. Another study reported the efficiency of ML models and found 93% accuracy for NB and SVM, 94% for RF and AdaBoost, and 95% for KNN models [54]. The AdaBoost model optimized in this study showed an advantage over all other models by detecting threats with high accuracy, of greater than 0.95, for both normal and anomalous data identification. AdaBoost performs exceptionally well at differentiating between true and false positives at various thresholds, as indicated by an ROC area of 0.98. This suggests the greater reliability of the model for different sensitivities in cyber-security scenarios. The model training and validation score demonstrates its robustness. The AdaBoost model is well scaled to handle large datasets, and therefore can be effectively used in real-world security applications. The previous literature on the significance of machine learning in cyber-security is summarized in Table 2 below.

## 3. Methodology

There are various phases of the methodology for distributed and scalable machine learning-based systems that are used for proactive threat hunting in critical infrastructure. These phases of the methodology include data collection, architecture, data pre-processing, selection and training of the machine learning model, model validation, and performance evaluation of the models, as given in detail below. Every phase of the methodology is designed for unique challenges and critical infrastructure to evaluate the real-time threats.

### 3.1. Data Collection 

For this study, real-time open-source data were collected from websites including Hugging face, GitHub, and Kaggle. This dataset mimics the real-time cyber-security data, containing both normal and anomalous data. The anomalous data are based on banking transactions for unauthorized access, SWIFT attacks, and man-in-the-middle (MITM) attacks. These attacks are common in critical infrastructure; therefore, machine learning models were trained and tested on this dataset to determine their effectiveness in real-world critical infrastructure applications. These real-time data streams are handled by efficient data pipelines, which also use safe storage techniques and encryption to guarantee data integrity and secrecy. 

### 3.2. Experimental Setup

On a Windows 10 PC with a GeForce GTX 980 GPU (NVIDIA, Santa Clara, CA, USA) and an Intel 2.20 GHz CPU (Intel, Santa Clara, CA, USA), models were trained utilizing MATLAB. Google and NVIDIA collaborated to provide GPUs and CPUs in order to increase processing performance. Python 3.8 was used for this investigation, and Google Colab, which has 16 gigabytes of RAM, was used to conduct the experimentations. Google Colab is a useful tool for executing code and carrying out tests because of its intuitive and smooth integration with Python. Larger datasets and memory-intensive operations are supported by the platform’s considerable RAM capacity, which makes more thorough investigations possible. 

### 3.3. Data Pre-Processing

In next phase, data pre-processing was carried out to convert data into an appropriate format before being used as input. First, data cleaning was conducted to handle the missing values in the dataset. A large number of missing values were identified, which were then filled by the technique of imputation and removal. Some missing values were handled by estimating the mean of the observed values. However, a large number of missing values were difficult to handle; therefore, these were omitted from the dataset by deleting the rows and columns containing null values. Then the data were further processed to eliminate the duplicate and irrelevant data that skew the results. The exploratory data analysis was also conducted by analyzing the pair-plot and matrix diagram to analyze the patterns of variables with respect to each other. Moreover, outliers can distort the training dataset; therefore, these were also detected and removed, which improved the accuracy of the models. The isolation forest model was utilized to detect and further handle the outliers in the dataset. Data encryption was used to ensure the security of the data and make it easier for various ML models to work together while protecting sensitive information from online threats.

### 3.4. Machine Learning Models

A variety of machine learning (ML) models, such as random forest (RF), support vector machine (SVM), multi-layer perceptron (MLP), AdaBoost, and hybrid models, were tested to identify the anomalies. Using labeled datasets and historical data, the ML models were trained. Supervised learning was used for recognized threat patterns, while unsupervised learning was used for anomaly detection. 

### 3.5. Model Evaluation

To improve model performance, cross-validation and hyperparameter adjustment were carried out. The system enables periodic retraining of models using updated datasets, and facilitates continuous learning from fresh data to respond to evolving threats. Accuracy, precision, recall, F1-score, and ROC curves were among the assessment matrices that were used to assess each model’s effectiveness. The formulas for calculating the metrics are given below in equations. The methodology flowchart is displayed in Figure 1 below.
(1)$$Precision=\frac{True\;Positive}{True\;Positive+False\;Positive}$$
(2)$$Accuracy=\frac{Number\;of\;correct\;prediction}{Total\;number\;of\;Predictions}$$
(3)$$Recall=\frac{True\;Positive}{True\;positive+False\;Negative}$$
(4)$$F1\;score=\frac{Precision\;*\;Recall}{Precision+Recall}$$

## 4. Results and Discussion

The findings of this study consist of important domains, including experimental data analysis, data preparation, and the use of machine learning models for anomaly detection and threat hunting, which are presented in Section 4. Exploratory data analysis makes the dataset’s primary structures and models visible. To ensure data quality and applicability for model training, pre-processing techniques were used. In addition, this section defines how different machine learning models were applied to identify possible threats and offers insights into their efficacy, as well as study implications. 

### 4.1. Exploratory Data Analysis (EDA)

Exploratory data analysis was a significant phase of this study that revealed the patterns, trends, and types of data used for machine learning models. The statistical evaluation of the dataset was conducted and is given in Table 3. 

Moreover, the pair-plot of the dataset (Figure 2) provided the visual representation of the relationship between pairs of features in the dataset. Plots of this kind are especially helpful in determining probable patterns, correlations, and distributions between various variables. The figure indicates that the card_present_flag plot shows no significant link with other features. When the card is present, the transaction appears to be more dispersed throughout a larger range of amounts, which are rather concentrated at smaller amounts. The distribution of the balance variable is right-skewed, indicating that while most people have smaller balances, a small percentage have noticeably higher amounts. The relationship between balance and other variables is more complex in the scatter plot. The distribution of the age variable is slightly skewed to the left, showing a higher percentage of younger people, whereas the amount variable is heavily skewed to the right, indicating a few high-value transactions. These skewed distributions are clear indications of the presence of outliers. Potential outliers are highlighted by the scatter plots, particularly in terms of amount and balance. These outliers can require attention during data pre-processing, or they might be areas of interest for additional investigation. Plots with concentrated points in specific places may indicate common trends or clusters that could be helpful for analyzing customer behavior or developing prediction models.

### 4.2. Data Pre-Processing

Data pre-processing is a significant step as it influences a model’s quality and effectiveness. Data are arranged into a clean and usable format that allows models to learn from the data more efficiently. A correlation matrix of all categorical columns, as shown in Figure 3, was analyzed for feature extraction. Moreover, the detection of outliers for their removal in data pre-processing is also a crucial step to improve the model’s performance. These outliers can skew the results and negatively influence the machine learning models. 

Figure 4 indicates the number of outliers detected over time. The number of outliers found is considerable in the beginning and peaks at about 20. This suggests that the data may have experienced some initial instability or noise. Over time, there are noticeable swings in the number of outliers, with both larger and lower outlier counts. This variability shows that throughout time, the nature or quality of the data may vary. The number of outliers jumps at a few significant points in the data, including the beginning and the end of the period. These peaks might point to occurrences, irregularities, or notable modifications in the underlying process that produced the data during the data collection process.

### 4.3. Threat Detection by ML Models

The next step is the threat detection by utilizing ML classifiers. Various ML models were applied to identify threats or anomalies in the real-time dataset of critical infrastructure. These include RF, SVM, MLP, AdaBoost, and hybrid models, which were assessed for threat hunting, and their performance was compared. These reconstruction-based models provide greater sensitivity, enabling more threat detection. The performance of these models was analyzed by utilizing the confusion matrix, ROC curve, and precision–recall curve. The details of all these models are as follows.

#### 4.3.1. Random Forest (RF) Model

The random forest model was tested on the dataset and the following results were obtained, as shown in Figure 5. A value of 0 in the figure indicates a negative class (normal data), while 1 denotes a positive class (anomalies). The confusion matrix shows that this model correctly identified 132 threats in the dataset. Moreover, 2069 were instances where this model correctly determined the absence of a threat. However, 31 false positive and 84 false negative instances were errors determined in this model while searching for an intrusion in the data framework. 

The ROC and precision–recall curves of the RF model are shown in Figure 6 below. The ROC curve plots the true positive rate against the false positive rate across various thresholds. The area under the ROC curve determines the degree of the model’s quality and discriminates whether the model satisfies the specific conditions or not. The area greater than 0.98 indicates excellent performance of RF in terms of threat identification. However, the F1-score is about 0.7, which indicates accurate detection of threats at first, but precision is lost by identifying more false positives. The model is quite good at differentiating threats and normal data, as evidenced by its high AUC value. The F1-score exhibits reasonable balance but precision becomes compromised as recall rises. The detailed classification report of the evaluation metrics is given in Table 4. This table indicates greater precision, of 0.960, for normal data identification and 0.809 for anomaly detection. Moreover, high overall accuracy of 0.950 is observed for this model. Overall, these measures collectively suggests that the RF model performed well for threat hunting, with a robust ability to minimize false positives. 

#### 4.3.2. Support Vector Machine (SVM) Model

The comprehensive assessment of the model’s accuracy and reliability was also conducted using a confusion matrix (Figure 7). A significant number of normal data (2092) were correctly identified by SVM model, indicating its good performance in identifying normal instances. Very few errors (8) were observed as false positives. However, the model failed to identify a considerable number of actual threats (188), which is a critical concern for threat detection application. This model successfully captured a small number of threats (28), highlighting the need for improvement. 

The efficiency of the SVM model was also analyzed using ROC and precision–recall curves (Figure 8). The area under the curve (ROC) of the SVM model is 0.82, suggesting that the model can distinguish between normal and anomalous cases with reasonable accuracy. The ROC curve’s shape indicates that, across a range of thresholds, the model maintains a high true positive rate while limiting the false positive rate. The precision–recall curve plot is also essential where detecting the anomalies is crucial. The maximum F1-score achieved by the model is 0.22. This value reflects the trade-off between precision and recall, emphasizing the difficulty of detecting anomalies. The model achieves excellent precision at first, but as recall increases, it decreases. This shows that although the model may correctly detect some abnormalities, a higher proportion of true positives is accompanied by a higher proportion of false positives. The curve’s downward trend suggests that as the model tried to capture more true threats (high recall), the proportion of false positives also increased, reducing precision. Table 5 indicates the efficiency of the model in terms of evaluation metrics. This model exhibited high precision (0.917) for normal data detection, and low precision (0.77) for abnormal data, with 0.915 accuracy.

#### 4.3.3. Multi-Layer Perceptron (MLP) Model

The MLP model indicated efficient detection of both non-threat and threat data but also showed less precision for identification. Its confusion matrix (Figure 9) demonstrates a greater number of instances, 1172 and 171, for true negative and positive data in the data framework, while a large number of instances (928) were misclassified as false threats. Few of the threats (45) failed to be detected by the model. This indicates that the model has the ability to identify intrusion but also could lead to false alarms during an attack situation. 

The ROC and precision–recall curves together provide a nuanced understanding of the MLP model efficiency in terms of intrusion detection. The ROC area under the curve of 0.76 exhibits the moderate performance of the model (Figure 10). The shape of the curve suggests that this model sustains a comparatively high true positive rate at the expense of a greater false positive rate. Similarly, the low F1-score of 0.26 illustrates the difficulties in striking a balance between recall and precision. The detail of the classification report and accuracy of the MLP model are given in Table 6. This indicates a high precision of 0.96 for normal data and a very low precision of 0.155 for abnormal data identification. The overall accuracy of MLP model was found to be 0.579.

#### 4.3.4. AdaBoost Model

The confusion matrix of AdaBoost model is shown in Figure 11. The model’s excellent recognition of non-anomalous data is demonstrated by its exact identification of 2071 normal cases (class 0). It is important for practical anomaly detection systems to minimize disruptions caused by false alerts, and the low frequency of false positives (29) shows that the model does not frequently raise unnecessary alarms. The model properly found 146 real anomalies. This suggests a reasonable sensitivity level and shows that the model can identify true abnormalities in the dataset. Seventy real abnormalities (70) were missed by the model, which mistakenly classified them as typical occurrences. Even though this number is small, it is nevertheless noteworthy since false negatives in anomaly detection can have detrimental effects. 

The area under the ROC curve of 0.98 signifies excellent performance, as shown in Figure 12. The curve’s proximity to the top right corner denotes an elevated true positive rate and a low false positive rate, highlighting the model’s robustness in recognizing anomalies and reducing false alarms. Similarly, the model achieves the maximum F1-score of 0.75, as indicated by the precision–recall curve. The pattern of the curve shows that the model performs well up until recall increases to a point where precision begins to decline more sharply. The overall metrics calculated from these curves are given in Table 7, exhibiting the highest precision values of 0.96 and 0.83 for 0 and 1 classes, respectively. 

#### 4.3.5. Hybrid Model

Like all other models, the hybrid model also shows a strong ability to correctly recognize traffic, reducing the number of false alarms. This model also did not erroneously indicate normal traffic as attacks, as indicated by the comparatively low number of false positives. Even though 136 cases were correctly identified as cyber-attacks, there were still 80 cases when attacks went unnoticed (Figure 13). This indicates the lower effectiveness of the hybrid model as compared to the supervised AdaBoost model for cyber-intrusion detection.

Moreover, when evaluation curves of hybrid model were evaluated (Figure 14), the area for ROC curves was found to be 0.89. This signifies the strong performance of this model, with a higher true positive rate and fewer false positive values. This is also confirmed by the sharp decrease in precision in the precision–recall curve, indicating an F1-score of 0.70. The values of precision and accuracy for cyber-attack detection were calculated to be 0.786 and 0.949, respectively, as shown in Table 8. These indicators show that the hybrid model can identify cyber-attacks with a high degree of accuracy; nevertheless, further work needs to be done to improve overall precision and minimize false positives.

#### 4.3.6. Learning Curves for Classifiers

The learning curves for all the classifiers, as shown in Figure 15, indicate the model performance in terms of training and cross-validation scores as the number of training examples increases. Gaining a knowledge of these curves is crucial for recognizing bias–variance trade-offs, possible overfitting or underfitting problems, and learning behaviors of the models. The learning curve for RF demonstrates that the model fits the training data very well, possibly overfitting, as evidenced by the training score, which stays constantly high, close to 1.0. The validation score was constant between 0.92 and 0.90, indicating a strong extrapolation to unobserved data. Given the narrow difference between the cross-validation and training scores, it can be assumed that the random forest model operates well with little overfitting, resulting in high accuracy and good generalization. The learning curve of SVM suggests that the training score was first high but then decreased as the number of training examples increased, eventually stabilizing around 0.91. The stabilization of the validation score around 0.91 indicates consistent performance for the tested data. The SVM model appears to have good generalization with the data, as indicated by the convergence of both training and testing scores. The training score of MLP varies but typically remains high, between 0.9 and 0.95. Additionally, there is diversity in the cross-validation score, which typically ranges from 0.85 to 0.90, demonstrating the generalization performance. The MLP model may be overfitting to the training data, as indicated by the wider difference between the training and cross-validation scores, with oscillations showing sensitivity to various subsets of the data. A successful fit on the training data is indicated by the training score, which begins high and gradually declines before stabilizing at 0.96 to 0.98. Good performance on unseen data is indicated by the cross-validation score, which exhibits more variation but often hovers around 0.92 to 0.94. Although the AdaBoost model retains a rather excellent generalization performance, the difference between the training and cross-validation scores points to a possible modest overfitting of the model. The training score for the hybrid model remains constantly high, ranging from 0.95 to 0.97, suggesting that the hybrid model provides a very good fit to the training set. Cross-validation scores are less consistent, ranging from 0.88 to 0.92. This score is a more accurate indicator of the model’s performance in actual settings. The validation stability indicates that the performance of the model is neither considerably enhanced nor diminished by the addition of new data. Variabilities in the model’s performance are indicated by fluctuations in the training and cross-validation scores. These variations may be brought about by the data’s nature or the model’s sensitivity to various subsets of the data. The cross-validation score exhibits a greater degree of fluctuation, suggesting possible discrepancies in the model’s ability to generalize.

### 4.4. Comparative Analysis

The comparative analysis of all models was also carried out in terms of evaluation metrics to determine the best optimized model for threat hunting. Firstly, the comparison of the main parameter of the area under the ROC curve was conducted to determine the efficiency of models with respect to others, as shown in Figure 16. As we discussed earlier, the ROC curve is regarded as an important parameter to identify the accuracy of the model. A value between 0.7 and 0.8 is regarded as acceptable, 0.8–0.9 is considered excellent, while above 0.9 is considered outstanding. The figure indicates the outstanding performance of RF and AdaBoost models in terms of the ROC curve, while the hybrid model shows slightly less performance, and a lower ROC value was observed with the MLP model.

The comparison of the precision, recall, and F1-score for anomalous (Figure 17a) and normal data (Figure 17b) detection, along with the accuracy of each model, is given in the form of bar graphs shown in Figure 17 below. This also indicates that the AdaBoost algorithm depicts highest precision and accuracy for anomaly detection. Moreover, the value for recall and accuracy is below 0.8 but still greater than that of all others. This exhibits the appreciable effectiveness of the AdaBoost model in differentiating the normal and anomalous data from a real-time dataset in cyber-security. This efficiency trend is followed by RF and hybrid models, which exhibit almost equal efficiency to that of the AdaBoost model in terms of these evaluation metrics. Moreover, the SVM model shows lower recall and F1-score for threat detection. Furthermore, the worst performance is demonstrated by the MLP model, with small precision and F1-score for cyber-threat detection in critical infrastructure. Figure 17b indicates evaluation metrics for normal data detection. This figure clearly shows that all models indicate reasonably high performance in terms of normal data detection without any errors, while the MLP algorithm also indicated small recall and F1-score for normalized data detection in the real-time dataset. By considering the learning curves, evaluation metrics, and ROC curve area, AdaBoost outperformed all other models and was found to be the most optimized for threat hunting. By analyzing all these metrics and ROC curve patterns, we concluded the following order of model performances in terms of cyber-threat detection: AdaBoost > RF > hybrid > SVM > MLP. 

## 5. Conclusions

Early anomaly detection in software-defined networking has an extensive impact on the network’s operational efficiency. The latest developments in ML aid in effective anomaly identification and improve service quality. Here, we investigated the use of RF, SVM, MLP, AdaBoost and hybrid machine learning models in tandem for identifying anomalies and offer a thorough overview of network architecture. Firstly, we talk about the limits of the current methods and the significance of identifying anomalies in contemporary networks. We outline their fundamental idea, possible uses, advantages, and drawbacks. Additionally, we included a thorough synopsis of those methods.

This study emphasizes the crucial role of ML in bolstering cyber-security for critical infrastructure.Random forest and AdaBoost models displayed exceptional performance, each with a 0.98 ROC area and overall accuracies of 95% and 95.7%, respectively.The hybrid model showed potential, with a 0.89 ROC area and 94.9% accuracy, although it requires improvement to lower false positives.ML models’ continuous learning capabilities ensure that they can adapt to new and emerging threats, enhancing the accuracy and speed of threat detection.Our work sheds light on how to build optimized autonomous models that can protect the system from sophisticated cyber-attacks. Future studies should try to replicate this study across a range of operational contexts and data variations in more general scenarios. We used learning curves to assess the model feasibility in terms of threat detection. Gaining a knowledge of these curves is crucial for recognizing bias–variance trade-offs, possible overfitting or underfitting problems, and learning behaviors of the models.

## Figures and Tables

**Figure 1 sensors-24-04888-f001:**
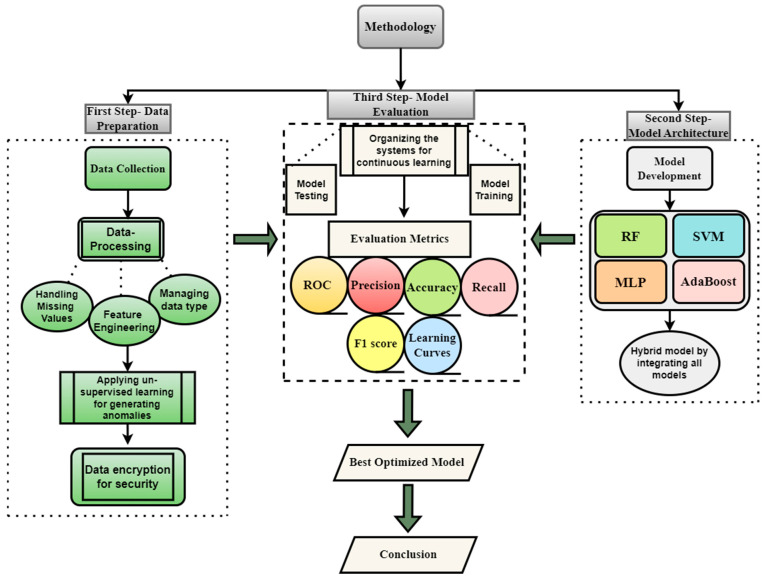
Flow chart of the methodology used for cyber-attack detections.

**Figure 2 sensors-24-04888-f002:**
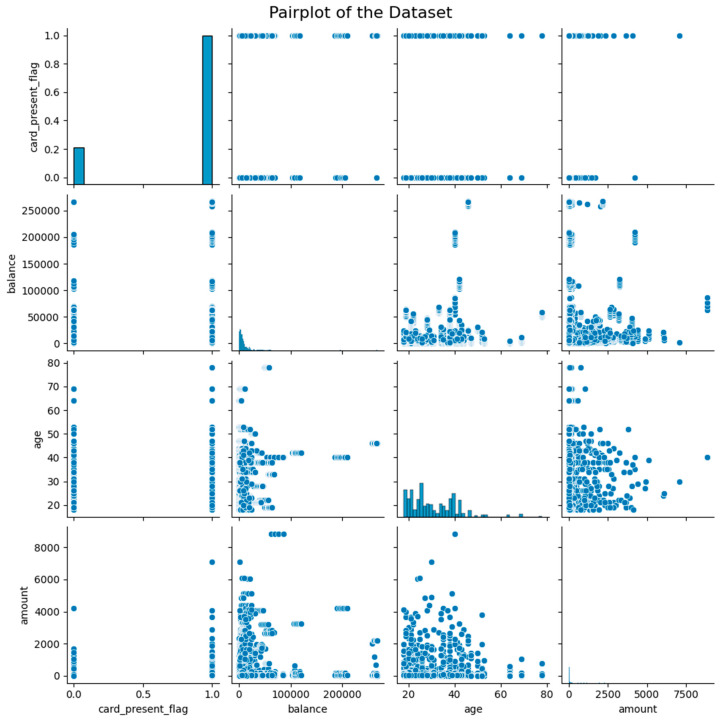
Pair-plots of the dataset.

**Figure 3 sensors-24-04888-f003:**
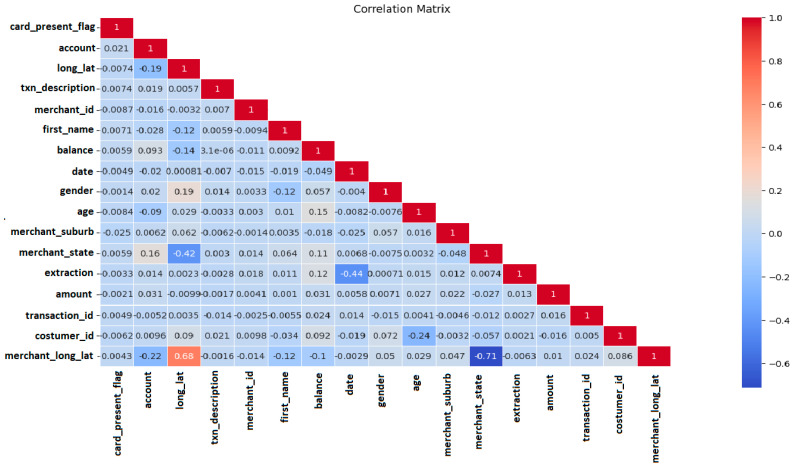
Correlation matrix of the dataset.

**Figure 4 sensors-24-04888-f004:**
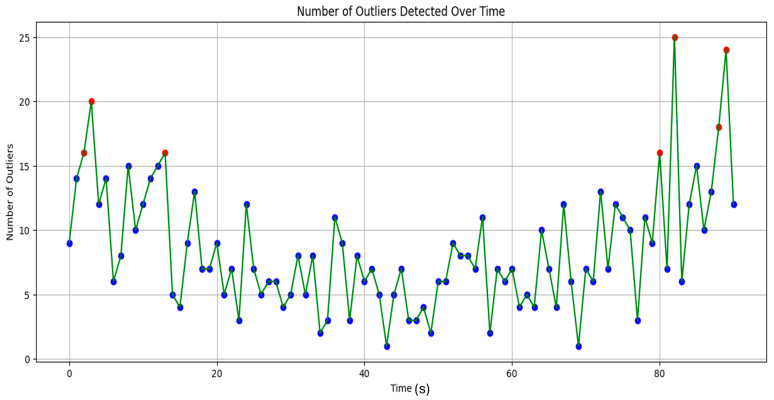
Detection of outliers in the dataset.

**Figure 5 sensors-24-04888-f005:**
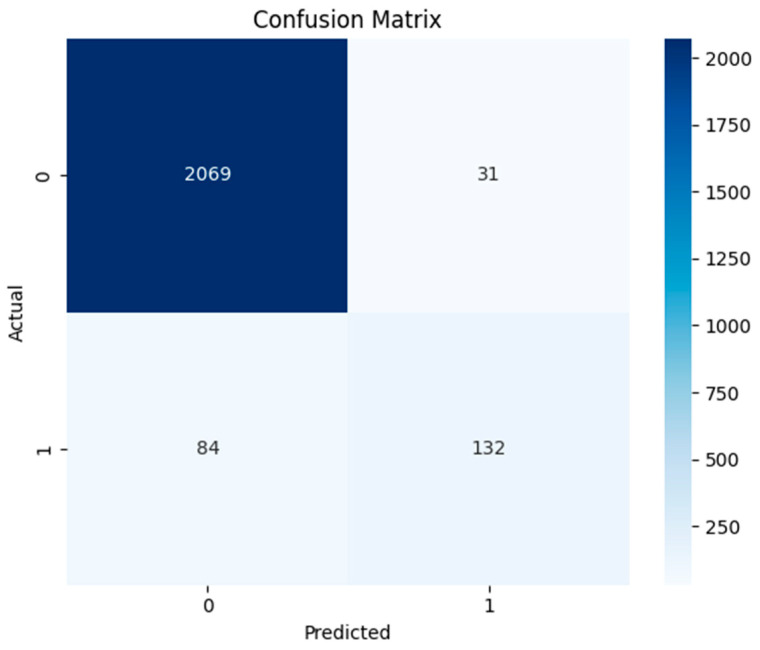
Confusion matrix of random forest model.

**Figure 6 sensors-24-04888-f006:**
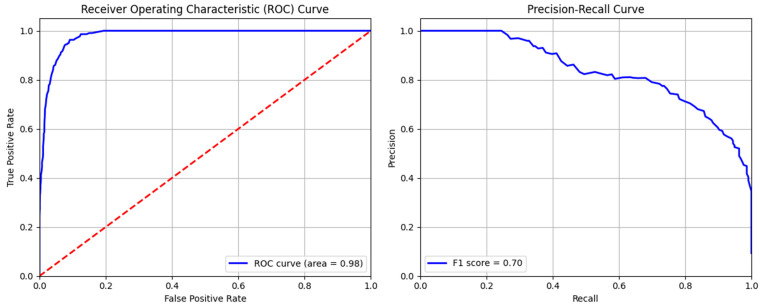
ROC and precision–recall curve of random forest model.

**Figure 7 sensors-24-04888-f007:**
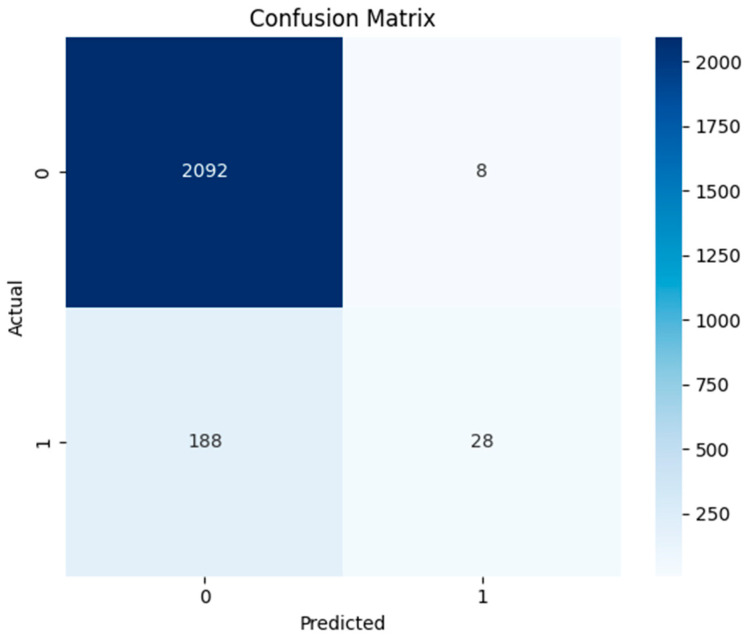
Confusion matrix of support vector machine model.

**Figure 8 sensors-24-04888-f008:**
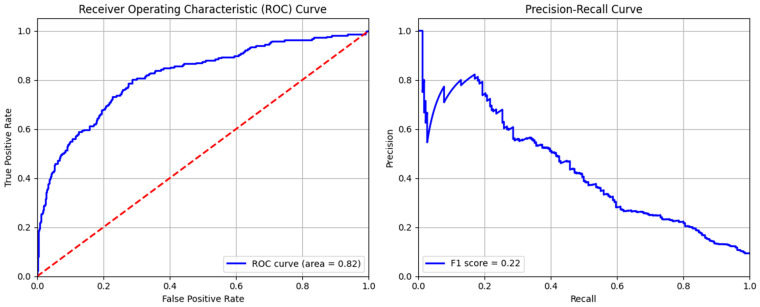
ROC and precision–recall curve of support vector machine model.

**Figure 9 sensors-24-04888-f009:**
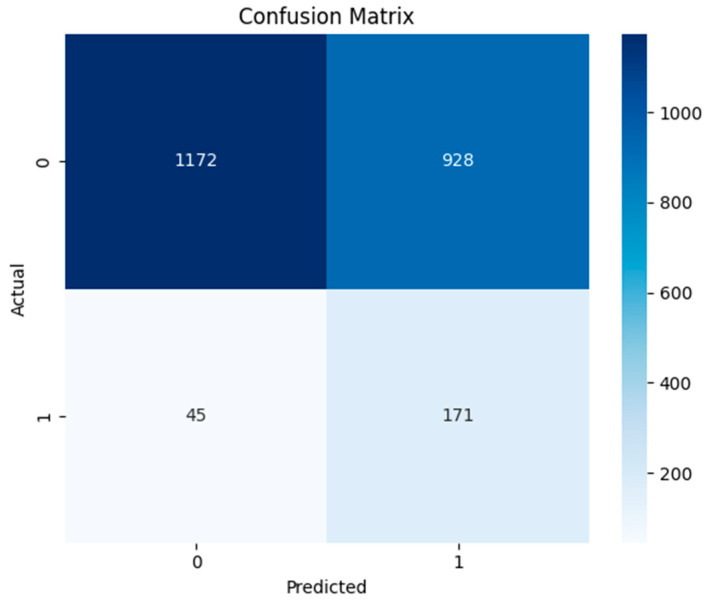
Confusion matrix of multi-layer perceptron model.

**Figure 10 sensors-24-04888-f010:**
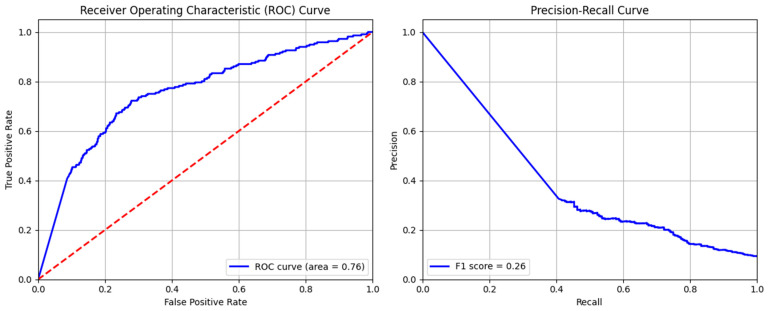
ROC and precision–recall curves of multi-layer perceptron model.

**Figure 11 sensors-24-04888-f011:**
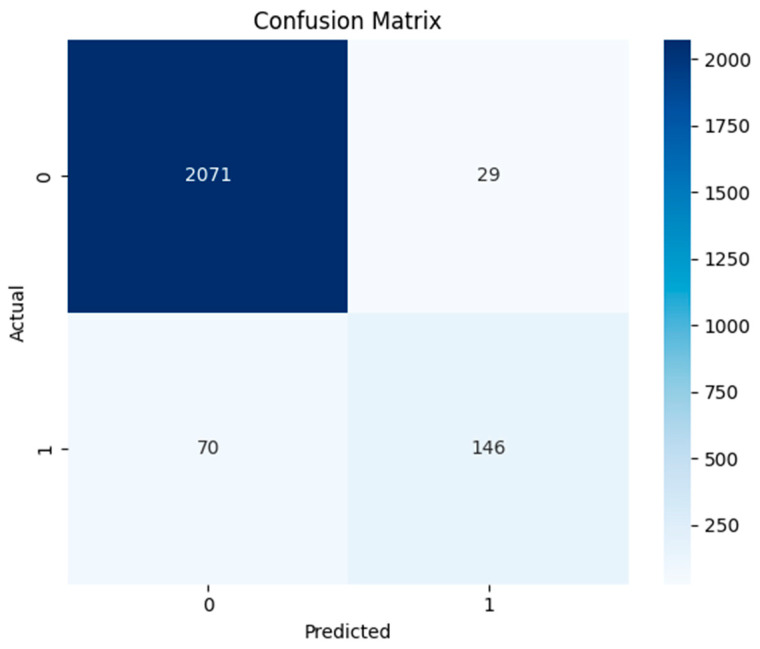
Confusion matrix of AdaBoost model.

**Figure 12 sensors-24-04888-f012:**
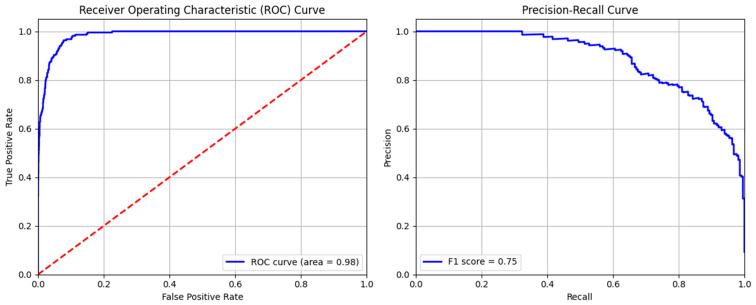
ROC and precision–recall curves of AdaBoost model.

**Figure 13 sensors-24-04888-f013:**
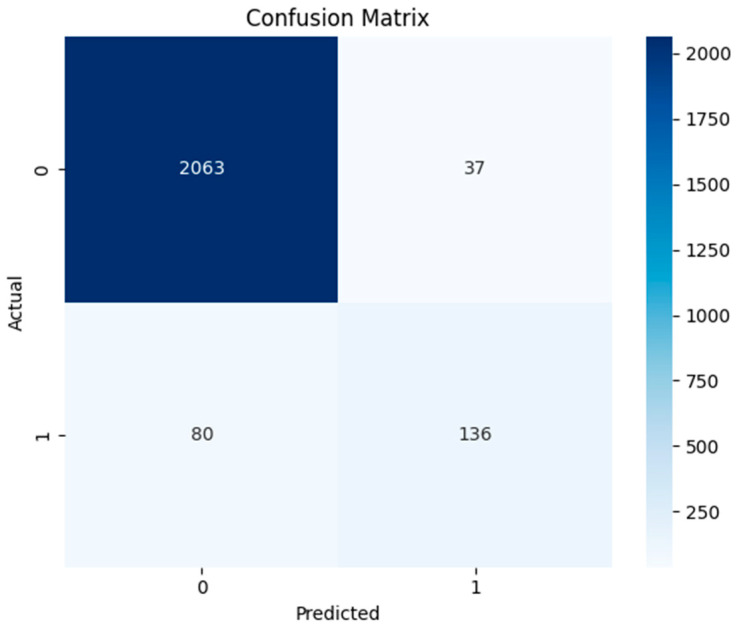
Confusion matrix of hybrid model.

**Figure 14 sensors-24-04888-f014:**
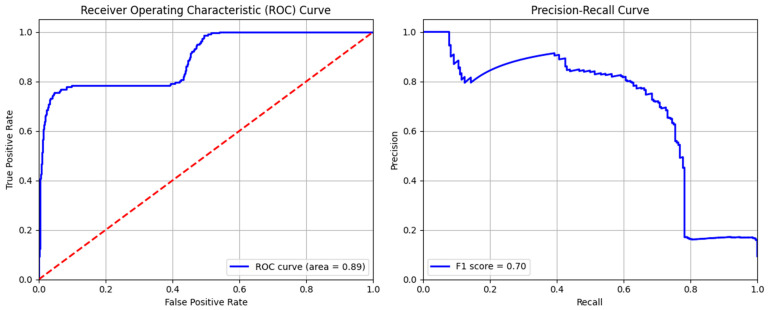
ROC and precision–recall curves of hybrid model.

**Figure 15 sensors-24-04888-f015:**
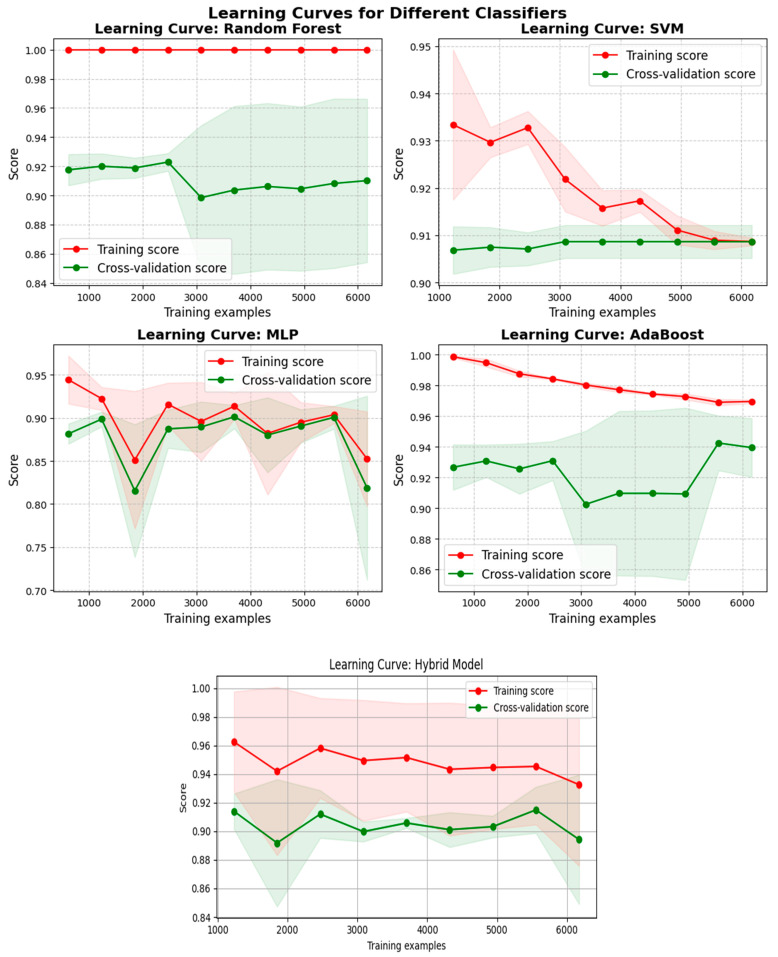
Learning curves of all classifiers.

**Figure 16 sensors-24-04888-f016:**
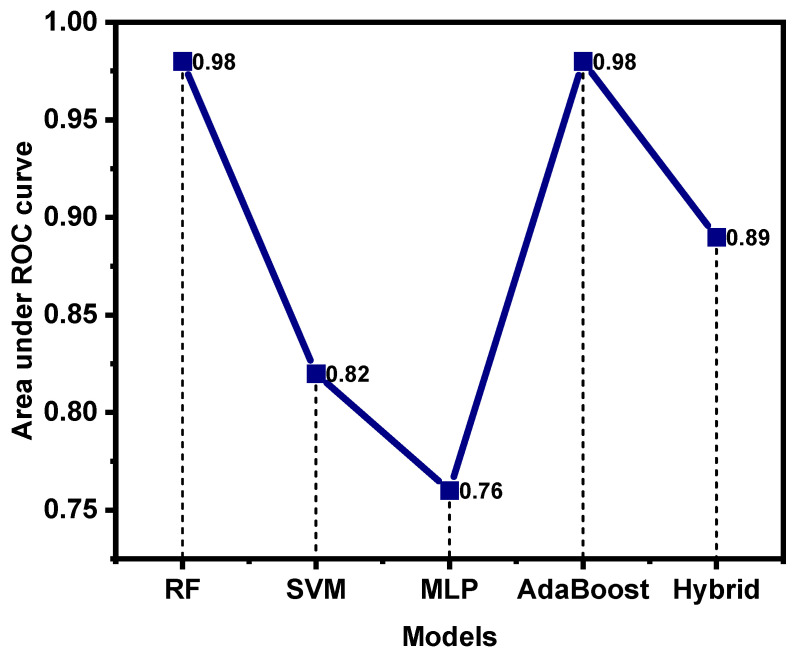
Comparison of area under ROC curve of all models.

**Figure 17 sensors-24-04888-f017:**
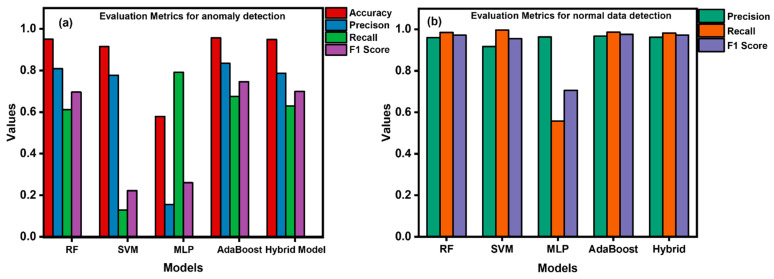
Comparison of evaluation metrics of (**a**) anomaly detection and (**b**) normal data detection for all models.

**Table 1 sensors-24-04888-t001:** Proactive techniques used.

Problem	Methodology Used	Findings	Reference
**Risk mitigations and proactive threats**	Reactive defense measured	Highlights threats and risk mitigations solutions	[35]
**Cyber-attacks**	Early response/proactive response	Changes active to proactive security system	Both AI and ML used in future [36]
**Data breaches in business industries**	IoT used	Recommend legacy system	[37]
**Cyber-attacks**	Artificial Intelligence based	Enhanced decision support system	[38]
**Cyber-attacks**	Bayesian long-short term memory	Holistic approach covering 42 attacks across 36 countries.	[39]
**Distributed reflection denial of service (DRDoS)**	Proposed proactive feature selection model (PFS)	PFS shows better accuracy for DRDoS detection	[40]
**Destructive cyber-threats**	Neural network-based machine and deep learning approach	Proactive cyber-threat detection with greater accuracy and prediction	[41]

**Table 2 sensors-24-04888-t002:** Significance of machine learning in cyber-security.

Techniques	Problems	Findings	Advantages	References
**IF and convolutional neural network (CNN) models**	Hybrid cyber-attack detections	Detected the maximum attacks with maximum accuracy	Proposed hybrid model of reasonable efficiency but lacks comparison	[64]
**NB, RF, and J48 model**	To detect DDoS attacks	Random forest model is more accurate than other models	Efficient attack detection in SCADA system but limited to one attack type	[65]
**DT model**	User-centric security and fake data identification for IoT-based critical infrastructure	To find theoretical and experimental solutions that solve security issues	Secure channel by decision tree in IoT security. Lacks comparative study	[66]
**RF, DT, SVM, perceptron, and Gaussian NB classifier**	Secure data dissemination architecture	Accurate, secure, and reliable architecture for IoT-based critical infrastructure	Efficient cyber-security in critical infrastructure but exhibited less accuracy	[67]
**RF, SVM, MLP, AdaBoost and hybrid model**	Cyber-threat detection from real-time dataset	Efficient threat hunting with high accuracy and precision; AdaBoost outperformed all models	All models performed well for cyber-threat detection. Comparative analysis shows high effectiveness of AdaBoost, RF, and hybrid models for real-world application.	This study

**Table 3 sensors-24-04888-t003:** Statistical evaluation of the dataset.

Parameter	Card Present Flag	Balance	Age	Amount
Count	7717	12,043	12,043	12,043
Mean	0.80	14,704	30.6	188
Std	0.39	31,504	10.0	593
Minimum	0.00	0.24	18.0	0.10
50%	1.00	6432	28.0	29.0
Maximum	1.00	26,713	78.0	8836

**Table 4 sensors-24-04888-t004:** Evaluation metrics for random forest models.

	Precision	Recall	F1-Score	Support
**0**	0.960	0.985	0.972	2100
**1**	0.809	0.611	0.696	216
**Accuracy**	0.950	0.950	0.950	0.950
**macro avg**	0.885	0.798	0.834	2316
**Weighted avg**	0.946	0.950	0.947	2316

**Table 5 sensors-24-04888-t005:** Evaluation metrics for support vector machine models.

	Precision	Recall	F1-Score	Support
**0**	0.917	0.996	0.955	2100
**1**	0.777	0.129	0.222	216
**Accuracy**	0.915	0.915	0.915	0.915
**macro avg**	0.847	0.562	0.588	2316
**Weighted avg**	0.904	0.915	0.886	2316

**Table 6 sensors-24-04888-t006:** Evaluation metrics for multi-layer perception model.

	Precision	Recall	F1-Score	Support
**0**	0.963	0.558	0.706	2100
**1**	0.155	0.791	0.260	216
**accuracy**	0.579	0.579	0.579	0.579
**macro avg**	0.559	0.674	0.483	2316
**Weighted avg**	0.887	0.579	0.665	2316

**Table 7 sensors-24-04888-t007:** Evaluation metrics for AdaBoost models.

	Precision	Recall	F1-Score	Support
**0**	0.967	0.986	0.976	2100
**1**	0.834	0.675	0.746	216
**Accuracy**	0.957	0.957	0.957	0.957
**macro avg**	0.900	0.831	0.861	2316
**Weighted avg**	0.954	0.957	0.955	2316

**Table 8 sensors-24-04888-t008:** Evaluation metrics for hybrid model.

	Precision	Recall	F1-Score	Support
**0**	0.962	0.982	0.972	2100
**1**	0.786	0.629	0.699	216
**Accuracy**	0.949	0.949	0.949	0.949
**macro avg**	0.874	0.806	0.835	2316
**Weighted avg**	0.946	0.949	0.946	2316

## Data Availability

Data is available on request.

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
