# Peer review of "Proactive Threat Hunting in Critical Infrastructure Protection through Hybrid Machine Learning Algorithm Application"

_sensors, 2024, doi:10.3390/s24154888_

Round 1

Reviewer 1 Report

Comments and Suggestions for Authors

The following points are put forward for the authors: 1.The authors need to explain the methodology used for implementing the machine learning models and conducting the threat detection analysis in detail. 2.The authors need to make a comprehensive comparison with existing methods or approaches in proactive threat hunting for critical infrastructure protection. 3.The authors need to discuss the computational resources required for running the models and the scalability of the approach.

Author Response

Thank you for allowing us to submit a revised draft of our manuscript to Sensors, We appreciate the time and effort that the reviewers have invested in providing insightful feedback on our manuscript. Care has been taken to improve the work and address their concerns as per the specific comments below. We have been able to incorporate changes that reflected the majority of the reviewers’ suggestions. The document has underlined every change that has been suggested.

Here is a point-by-point response to the reviewer’s comments and concerns.

Comments from Reviewer 1

  • Comment 1: The authors need to explain the methodology used for implementing the machine learning models and conducting the threat detection analysis in detail.

Response: Thank you for your suggestion. We have now explained the detailed methodology used for implementing machine learning and threat detection. Page # 7, 8 lines 278-333.

  • Comment 2: The authors need to make a comprehensive comparison with existing methods or approaches in proactive threat hunting for critical infrastructure protection.

Response: We have now compared our study with the existing related literature as per the reviewer suggestion. Page 6, 7 line 252-270.

  • Comment 3: The authors need to discuss the computational resources required for running the models and the scalability of the approach.

Response: Thank you for pointing this out. The computational resources required for running the model has been now discussed in the manuscript. Page 7, line 283-291.

Reviewer 2 Report

Comments and Suggestions for Authors

The topic of automatically tracking threats is extremely relevant.

It should be noted that a large amount of illustrative material is presented, but there is a lack of clarity in the transitions between sections.

It should also be noted that there is no gain from Figures 16 and 17 when using the proposed hybrid model. It can be assumed that her gain lies in something else. Should it be shown more clearly why the proposed model is better than existing ones (e.g. AdaBoost)?

In Table 2, it is better to change the order of the columns, and the links can be combined with "Findings."

Table 3 should clarify rounding accuracy.

The description of Figure 2 contains general phrases about the purpose of pair-plot but does not analyze the patterns presented in the graphs.

In Figure 3, you can delete half of the values, since they are duplicated (leaving a triangular table).

Figure 4 starts in the negative region in time. There are not enough units of measurement on it.

Confusion matrices take up too much space. It may be better to present them in the form of tables.

Plans for future research are best indicated at the end of the conclusions (see lines 455 and 456).

The use of abbreviations is chaotic, for example, in the conclusions the abbreviation “machine learning (ML)” is suggested.

Author Response

Thank you for allowing us to submit a revised draft of our manuscript to Sensors, We appreciate the time and effort that the reviewers have invested in providing insightful feedback on our manuscript. Care has been taken to improve the work and address their concerns as per the specific comments below. We have been able to incorporate changes that reflected the majority of the reviewers’ suggestions. The document has underlined every change that has been suggested.

Here is a point-by-point response to the reviewer’s comments and concerns.

  • Comment 1: It should be noted that a large amount of illustrative material is presented, but there is a lack of clarity in the transitions between sections.

Response: We apologize for this mistake. We have now tried to clear the purpose of each section in a clear manner.

Comment 2: It should also be noted that there is no gain from Figures 16 and 17 when using the proposed hybrid model. It can be assumed that her gain lies in something else. Should it be shown more clearly why the proposed model is better than existing ones (e.g. AdaBoost)?

  • Response: By appreciating the author’s comment, the comparison of AdaBoost model with the existing ones has now been made in the manuscript. Page 6, line 252-260.

  • Comment 3: In Table 2, it is better to change the order of the columns, and the links can be combined with "Findings."

Response: Thank you for the suggestion. Table 2 has now been modified and linked with the findings of this study. Page 6, 7.

  • Comment 4: Table 3 should clarify rounding accuracy.

Response: Table 3 has now been clarified by updating their rounding accuracy. Page 9.

  • Comment 5: The description of Figure 2 contains general phrases about the purpose of pair-plot but does not analyze the patterns presented in the graphs.

Response: We appreciate your concern. We have now added the detailed description of pair plot that includes information regarding the patterns presented in graphs. Page 9, line 342-350.

  • Comment 6: In Figure 3, you can delete half of the values, since they are duplicated (leaving a triangular table).

Response: We have now updated Figure 3 as per the reviewer suggestion. Page 11, Figure 3.

  • Comment 7: Figure 4 starts in the negative region in time. There are not enough units of measurement on it.

Response: Thank you for pointing this out. We have also updated Figure 4 with all unit of measurements on it. Page 11.

  • Comment 8: Confusion matrices take up too much space. It may be better to present them in the form of tables.

Response: We appreciate your concern. As each confusion matrix contains very small amount of data and it is not feasible to generate table of such small data of each confusion matrix. Moreover, we have also discussed each model separately to avoid any misunderstanding so the Table cannot be merge at a single place also. We tried to keep the size of all matrixes minimum so that they would not occupy much space as before.  

Comment 9: Plans for future research are best indicated at the end of the conclusions (see lines 455 and 456).

  • Response: We agree. The future prospects are now discussed at the end of conclusion along with the suggested lines. Page 21, line 596-602.

  • Comment 10: The use of abbreviations is chaotic, for example, in the conclusions the abbreviation “machine learning (ML)” is suggested.

Response: We apologize for this mistake. The abbreviations have now only been mentioned with their full terms when they were first time used.

Round 2

Reviewer 1 Report

Comments and Suggestions for Authors

This paper explores the use of various machine learning models to improve threat detection and enhance cybersecurity measures in critical infrastructure.

The following points are put forward for the authors:

1.The authors need to describe in detail the missing value treatment used.

2.The authors need to specify how the datasets used relate to the types of attacks on critical infrastructure.

3.The authors need to specify what types of attacks the proposed method can effectively detect against various types of attacks.

Author Response

Thank you for allowing us to submit a revised draft of our manuscript to Sensors, We appreciate the time and effort that the reviewers have invested in providing insightful feedback on our manuscript. Care has been taken to improve the work and address their concerns as per the specific comments below. We have been able to incorporate changes that reflected the majority of the reviewers’ suggestions. The document has underlined every change that has been suggested.

Here is a point-by-point response to the reviewer’s comments and concerns.

Reviewer Comments:
Comment 1.The authors need to describe in detail the missing value treatment used.
Response: By appreciating the author’s comment, the method used to handle the missing values
is discussed in manuscript. Page 7, Line 300-304.

Comment 2.The authors need to specify how the datasets used relate to the types of attacks on
critical infrastructure.
Response: Thank you for your suggestion. We have now discussed the specifications of the
dataset that how it is related to critical infrastructure. Page 7, line 279-287.

Comment 3.The authors need to specify what types of attacks the proposed method can
effectively detect against various types of attacks.
Response: The type of attacks in the dataset that the proposed models detected have now been
specified in the data collection of the manuscript. Page 7, line 281-285.